# Incidence and survival in laryngeal and lung cancers in Finland and Sweden through a half century

**Anni Koskinen[1], Otto Hemminki[2], Asta Försti[3,4], Kari Hemminki[5,6]***

**1** Department of Otorhinolaryngology, Head and Neck Surgery, Helsinki University Hospital and University of Helsinki, Helsinki, Finland, **2** Department of Urology, Helsinki University Hospital and University of Helsinki, Helsinki, Finland, **3** Hopp Children's Cancer Center (KiTZ), Heidelberg, Germany, **4** Division of Pediatric Neurooncology, German Cancer Research Center (DKFZ), German Cancer Consortium (DKTK), Heidelberg, Germany, **5** Biomedical Center, Faculty of Medicine and Biomedical Center in Pilsen, Charles University in Prague, Pilsen, Czech Republic, **6** Division of Cancer Epidemiology, German Cancer Research Center (DKFZ), Heidelberg, Germany

* k.hemminki@dkfz.de

## Abstract

Global survival studies have shown favorable development in most cancers but few studies have considered laryngeal cancer, particularly over extended periods or in populations for which medical care is essentially free of charge. We analyzed laryngeal and lung cancer incidence and survival in Finland (FI) and Sweden (SE) over a 50-year period (1970–2019) using data and statistical tools from the Nordcan database. Laryngeal cancer reached an incidence maximum in FI men in 1965, which in SE men occurred over 10 years later and peaking at 42% of the FI maximum. The FI incidence halved in 20 years while halving of the SE rate took almost twice as long. At maximum the male rate exceeded the female rate 20 times in FI and 10 times in SE. Incidence rates for lung cancer were approximately 10 times higher than those for laryngeal cancer, and they peaked 5 to 10 years after laryngeal cancer in both countries. The female lung cancer rates increased through the follow-up time but laryngeal cancer rates were relatively stable. Relative 1-year survival data for laryngeal cancer remained at around 85% through 50 years, and 5-year survival lagged constantly around 65%. For lung cancer 1-year survival improved and reached about 50% by 2019. Even 5-year survival improved reaching 20 to 30%, except for FI men. Incidence rates for laryngeal and lung cancers have drastically decreased in FI and SE men parallel to reduced smoking prevalence. In females, rates have clearly increased in lung but not in FI laryngeal cancer. This finding warrants further investigations into possible contributing factors, other than smoking. Survival in laryngeal cancer has not improved compared to the positive development in lung cancer. Historical smoking prevalence was unrelated of survival trends. As long-term survival in these cancers remains discouraging, the most efficient way to fight them is to target the main cause and promote non-smoking.

**Funding:** Supported by the European Union's Horizon 2020 research and innovation programme, grant No 856620. The funder did not influence the conducted research. The funder had no role in study design, data collection and analysis, decision to publish, or preparation of the manuscript.

**Competing interests:** The authors have declared that no competing interests exist.

## Introduction

Smoking is a preeminent risk factor for laryngeal and lung cancers. Relative risks for tobacco-related lung cancer are of the order of 10 to 20 in active cigarette smokers compared to non-smokers, and they remain at levels of 3 to 5 after 20 years of quitting smoking [1, 2]. For laryngeal cancer, smoking-related risk is at the same level but relative reduction of risk after quitting is faster than what is known in lung cancer, and in 10 years the risk may reach the level of non-smokers [1]. According to analysis of population attributable fraction (PAF) for lung cancer in UK for 2015, 74.1% was assigned to smoking in men and 70.2% in women [3]. In laryngeal cancer the percentages were 64.6 and 61.4. For lung cancer, other contributors for men were occupation and air pollution, which influenced also women. For laryngeal cancer, 23.8% was assigned to alcohol in men compared to 11.8% in women.

The incidence in laryngeal and lung cancer up to year 2003 have been higher in FI men than that in SE men while in female rates no large differences were observed [4]. Considering the importance of smoking as a risk factor for laryngeal and lung cancers, we need to put it in perspective of the present Finnish (FI) and Swedish (SE) populations. We have collected these data from different sources (www.pnlee.co.uk/ISS.htm) and summarized these elsewhere [5–7]. Shortly, FI men were among the heaviest European smokers after World War II but started to reduce the habit already in the 1950s. Among SE men, smoking prevalence was at a moderate level and reached the lowest level among European men at 15% by the early 2000s [6, 8]. The drop in smoking prevalence was also fast in Finnish men but it remained at 30% into the early 2000s. The smoking prevalence in Swedish women was around 30% in 1960 and it slowly decreased towards 20% in the early 2000s when it exceeded the male prevalence [8]. Among Finnish women the trend was opposite, starting at <5% in 1960 and ending up at the Swedish level of 20% in the early 2000s. Alcohol is an important risk factor for laryngeal cancer. The FI consumption figures have been higher than the SE ones and deaths due to alcoholic liver disease have differed even more, 4-fold, between FI and SE [9]. However the FI excess in consumption of alcohol over SE was relatively recent, taking place in the early 1970s [10].

Survival in cancer is the difference between incidence and mortality, and it is influenced by many other factors, including diagnostics, treatment and supporting care [11]. Smoking and continued smoking in patients treated for laryngeal and lung cancer have been shown to worsen survival [12–15]. The study on Nordic laryngeal and lung cancer patients up to year 2006 conclude that small increases in relative survival were achieved for lung cancer and male laryngeal cancer but not for female laryngeal cancer [4]. In a more recent Danish study, no improved overall 5-year survival for laryngeal cancer was observed between 1980 and 2014 [16]. Curative therapies for local and locoregional laryngeal cancer are surgery and radiation therapy, when possible, using minimally invasive resection and hypofractionated radiotherapy [17]. In more advanced tumors comprehensive evaluation is needed to choose the treatment modality. Radiotherapy, in combination with chemotherapy, often cisplatin, is commonly used, whereas surgery, usually total laryngectomy with or without neck dissection, is considered mostly for patients with non-functional larynx or as salvage therapy [18]. For lung cancer, surgery is the main therapeutic modality for early stages, and radio- and chemotherapy are used in advanced stages [19]. The more common non-small cell lung cancer is relatively resistant towards chemotherapy and radiation, whereby surgery is the treatment of choice. Small cell carcinoma may initially respond well to chemotherapy and radiation, but has usually metastasized before diagnosis, making surgery ineffective [19]. For both cancers immunotherapy is being used in recurrent or metastatic patients but as the current follow-up ended in 2019 this treatment had no impact on the results.

We report here result on survival in laryngeal and lung cancers in FI and SE over a period of 50 years. However, in order to interpret the results we also show data for incidence in these cancers. The neighboring counties FI and SE are historically and culturally related but have many differences in health care resources and development. SE has been an international example of high-level medical care and its economy has been prosperous since the World War Two. FI has been a poor neighbor, however, trying to catching up and use SE as a model in medical care, yet with constantly lower share of health care expenditure (www.macrotrends. net). Both countries have offered medical care practically free-of-charge providing an opportunity to view a 'real world' experience of medical outcomes. A further point relevant to these cancers is the extensive differences between smoking histories in these two countries.

## Methods

The data used originate from the Nordcan database 2.0 which is a compilation of data from the Nordic cancer registries [20]. These registries have had a long-standing collaboration and decades of joint publications with comparable data as presented in detail by Pukkala and coworkers [21]. The database was accessed at the International Agency for Cancer (IARC) website (https://nordcan.iarc.fr/en/database#bloc2 Coverage of cancers in the FI and SE cancer registries is generally considered high [21]. The SE cancer registry does not consider cancers in death notifications and some 4% cases may be missed because of this; an overall comparison of various health records showed that the coverage was over 90% [21, 22]. Comparability of diagnostics of a 50-year period may be an issue. The SE cancer registry used ICD-7 from the start of registration in 1958. When new codes have been taken to use, all diagnoses are additionally recoded in the ICD-7 system to maintain consistence [23]. FI uses a code conversion system to maintain consistency.

Data on FI and SE patients were extracted from Nordcan and the follow-up was extended until death, emigration or loss of follow-up or to the end of 2019. All data displayed and statistical analyses were carried out with the tools provided at the Nordcan site; 95% confidence intervals (95%CIs) were considered for statistical significance. For incidence data, the starting date was the earliest available, 1953 for FI, 1960 for SE. For age standardization the Nordic standard population was used. In Nordcan, the tool for incidence analysis by period ('Time trends, Incidence') was selected for analysis. Survival data for relative survival were available from 1970 onwards and the analysis was based on the cohort survival method for the first nine 5-year periods, and a hybrid analysis combining period and cohort survival in the last period 2015–2019, as detailed [24, 25]. Age groups 0 to 89 were considered, and for age-standardization the International Cancer Survival Standard was used. The FI and SE life tables were used to calculate the expected survival. Period-specific survival analysis was executed with tool 'Table by country and period', and the results were compiled from the relevant analyses.

The Nordcan site had no tool for the age, period, period (APC) analysis. However, birth cohort analysis was possible, and this was applied on to Finnish male data (with the largest case numbers); the low case numbers for laryngeal cancer in other populations did not allow a meaningful analysis.

In graphic presentation of incidence rates, lines were smoothed by the LOESS regression algorithm (bandwidth: 0.1 or 0.3). The figure indicates the proportion of points in the figure that influence each point.

## Results

In period 1967 to 2019, the patient numbers of laryngeal and lung cancers from the Nordcan database are shown in **Table 1.** Among men, patient numbers for lung cancer (over 94,000 for

**Table 1. Patient numbers and median diagnostic ages for laryngeal and lung cancer in Finland and Sweden, 1967–2019.**

|  | Cases | Median age, years |
|---|---|---|
| **Finland, men** |  |  |
| larynx | 6196 | 64 |
| lung | 94797 | 68 |
| **Finland, women** |  |  |
| larynx | 665 | 67 |
| lung | 23700 | 70 |
| **Sweden, men** |  |  |
| larynx | 8756 | 67 |
| lung | 94379 | 71 |
| **Sweden, women** |  |  |
| larynx | 1349 | 66 |
| lung | 52456 | 69 |

FI and SE) exceeded those for laryngeal cancer (6196 FI and 8756 for SE) by 10 to 15-fold, but for women the differences were 35 to 40-fold (lung cancer 23,700 FI and 52,456 SE compared to larynx cancer 665 FI and 1349 SE). Laryngeal cancer was diagnosed 3 to 4 years earlier than lung cancer.

Incidence rates for laryngeal and lung cancers in FI and SE men and women are shown in **Fig 1**. Laryngeal cancer reached an incidence maximum in FI men in 1965, which in SE men

**Table 2. Relative 1- and 5-year survival for laryngeal cancer in Finland and Sweden.**

| Period | 1-year survival % [95% CI] | | 5-year survival % [95% CI] | |
|---|---|---|---|---|
| **Men** | **Finland** | **Sweden** | **Finland** | **Sweden** |
| 1970–1974 | 84.4 [77.6–89.3] | 89.3 [85.8–92.0] | 52.9 [41.8–62.8] | 67.9 [61.4–73.6] |
| 1975–1979 | 82.3 [76.6–86.7] | 86.4 [82.9–89.1] | 55.9 [48.4–62.7] | 62.6 [57.3–67.4] |
| 1980–1984 | 85.2 [79.8–89.2] | 87.6 [84.5–90.1] | 58.0 [49.4–65.7] | 66.7 [61.6–71.3] |
| 1985–1989 | 82.9 [78.3–86.6] | 87.5 [84.6–89.8] | 58.6 [51.7–64.9] | 68.8 [64.0–73.0] |
| 1990–1994 | 85.0 [80.5–88.5] | 86.6 [83.5–89.1] | 59.7 [52.8–65.9] | 67.7 [62.8–72.0] |
| 1995–1999 | 85.0 [80.3–88.6] | 88.6 [85.6–91.1] | 62.4 [55.2–68.7] | 67.8 [62.4–72.5] |
| 2000–2004 | 82.1 [77.2–86.0] | 85.9 [82.8–88.4] | 59.7 [53.1–65.7] | 66.3 [61.5–70.6] |
| 2005–2009 | 83.0 [78.6–86.5] | 87.5 [84.5–89.9] | 60.0 [53.8–65.7] | 68.4 [63.6–72.6] |
| 2010–2014 | 85.6 [81.3–89.0] | 89.7 [86.9–91.9] | 59.2 [53.1–64.9] | **70.8** [65.7–75.2] |
| 2015–2019 | 83.6 [79.7–86.8] | 88.9 [86.2–91.1] | 59.6 [53.9–64.8] | **71.9** [67.2–76.0] |
| **Women** | **Finland** | **Sweden** | **Finland** | **Sweden** |
| 1970–1974 | 61.6 [34.6–80.1] | 89.4 [74.4–95.9] | 52.5 [29.0–71.5] | 75.7 [45.8–90.6] |
| 1975–1979 | - | 88.1 [77.6–93.9] | - | 67.6 [48.6–80.9] |
| 1980–1984 | 77.0 [60.8–87.1] | 90.5 [80.8–95.4] | 56.7 [37.9–71.8] | 57.9 [44.7–69.0] |
| 1985–1989 | 77.4 [62.4–87.0] | 89.3 [81.3–94.0] | 55.2 [39.5–68.4] | 73.6 [60.4–83.1] |
| 1990–1994 | 77.2 [60.8–87.4] | 90.0 [81.8–94.6] | 58.7 [41.3–72.5] | 69.0 [57.3–78.1] |
| 1995–1999 | - | 86.1 [77.5–91.6] | - | 70.0 [58.0–79.2] |
| 2000–2004 | - | 86.3 [79.2–91.1] | - | 58.6 [49.0–67.0] |
| 2005–2009 | 85.7 [72.4–92.9] | 80.8 [73.2–86.5] | 66.1 [51.0–77.5] | 61.4 [51.6–69.8] |
| 2010–2014 | - | 87.1 [79.8–91.9] | - | 64.4 [54.6–72.6] |
| 2015–2019 | 83.8 [73.8–90.2] | 84.5 [75.5–90.4] | 58.5 [40.8–72.6] | 59.3 [49.1–68.1] |

Bolding: 95%CI do not overlap. Bolding shows the country of higher survival %.

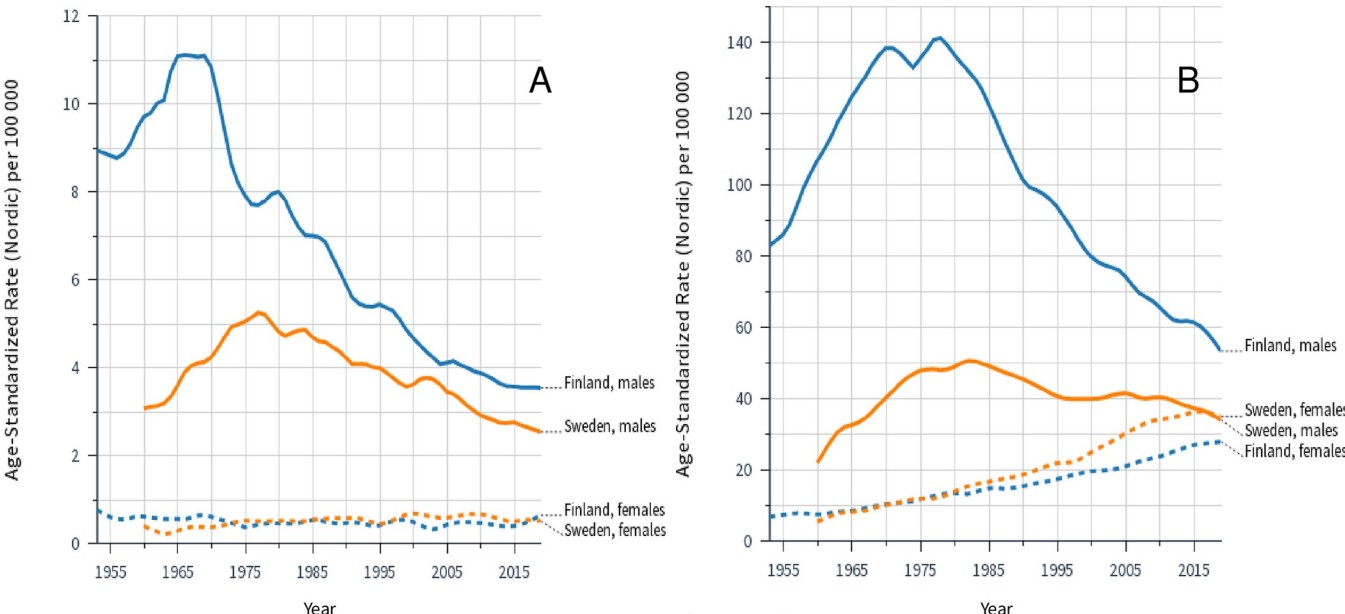

**Fig 1.** Age standardized incidence rates for laryngeal (A) and lung (B) cancers in Finland since 1954 and Sweden since 1960. Curves were plotted using smoothing bandwidth: 0.1. Note the different scales for incidence rates.

occurred more than 10 years later and peaking at 42% of the FI maximum (**Fig 1A**). The FI rate decreased fast, and halved in 20 years while halving of the SE rate took almost twice as long. Among women, laryngeal cancer incidence appeared stable over the follow-up period, with a small decline in FI from 0.7 to 0.6/100,000 and a small increase in SE from 0.3 to 0.4/100,000. Nevertheless, at maximum (1965) the FI male rate exceed the female rate 20 times; the maximal SE sex difference was 10-fold. Incidence rates for lung cancer were approximately 10 times higher than those for laryngeal cancer (**Fig 1B**). For males, the peak incidence of lung cancer occurred some 5 years later than that for laryngeal cancer, and the SE maximal lung cancer incidence was 30% of the FI maximum. The female lung cancer rates started to increase from the start of the follow up; the SE rate reached a maximum (36.8/100,000) in 2016 after a 7-fold increase; the FI rate continued to increase having reached a 5-fold increase in 2019 (28.0/100,000). The SE female rate exceeded the male rate towards the end of the follow-up period. The maximal sex difference in lung cancer was 15-fold in FI (1970) and 5-fold in SE (1975).

Birth cohort analysis of the Finnish male data showed that laryngeal cancer reached the incidence maximum 5 to 10 years earlier than lung cancer in each of age groups with the largest case numbers (**Fig 2**).

A more detailed analysis of female laryngeal cancer incidence showed that the rates were actually not stable (**Fig 3**). The FI rate constantly decreased to about 2010 and then modestly increased. The SE rate reached a maximum in 2005.

Relative survival data for laryngeal cancer are plotted in **Fig 4** (**A**, 1-year survival and **B**, 5-year survival), illustrating the constant SE rates and somewhat improving FI rates, particularly for women. While 1-year survival remained at around 85% through 50 years, 5-year survival lagged constantly about 20% units behind. FI female 1-year survival was initially far lower than for others but it caught up by year 2000.

Survival data for lung cancer is presented in **Fig 5**. In 1-year survival (**A**) FI men and women were initially leading at close to 40% but FI male survival hardly improved, compared

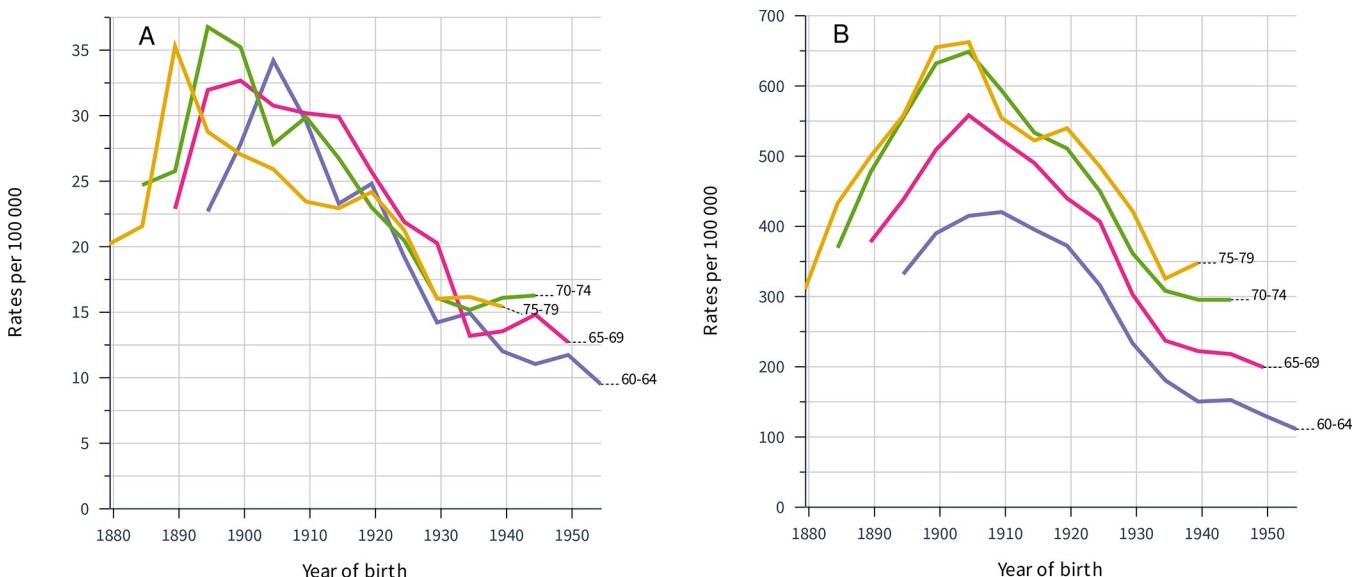

**Fig 2.** Birth cohort analysis of Finnish laryngeal (A) and lung (B) cancer incidence in the age groups with the largest case numbers for these cancers.

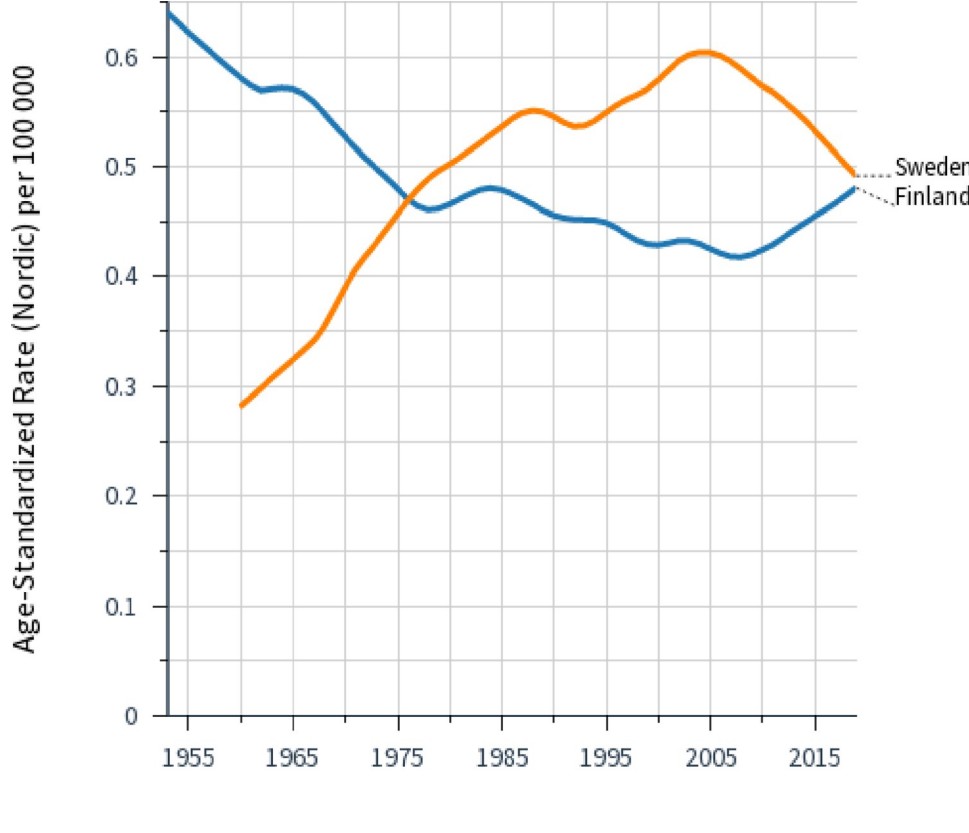

**Fig 3. Age standardized incidence rates for female laryngeal cancer in Finland since 1954 and Sweden since 1960.**
Curves were plotted using smoothing bandwidth: 0.3.

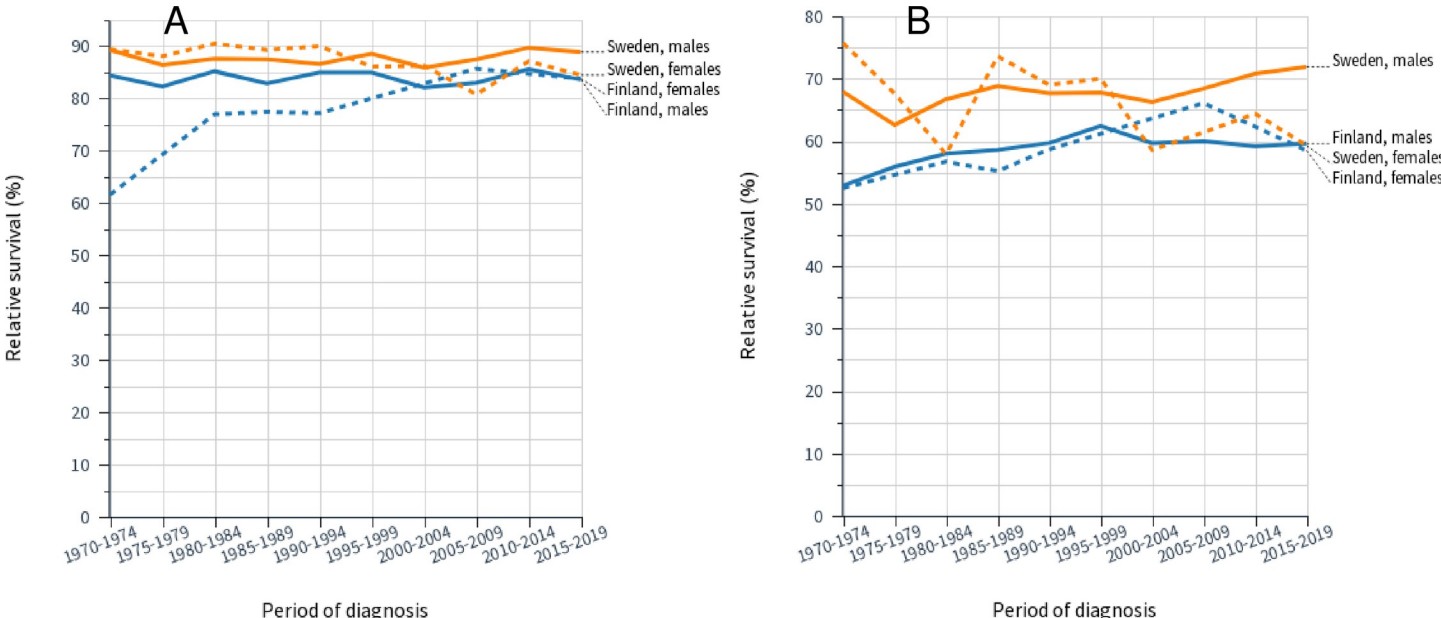

**Fig 4.** Relative 1-year (**A**) and 5-year (**B**) survival rates for laryngeal cancer in Finnish and Swedish men and women. The related 95%CIs for the data points are shown in Table 2.

to the other groups whose survival reached about 50% (SE women close to 60%) by 2019. Even 5-year survival (**B**) improved, especially after 2005, reaching over 20% for SE women but only 16% for FI men.

Relative 1-year and 5-year survival rates for laryngeal cancer for FI and SE men and women were analyzed in more detail in **Table 2**. The surprising feature about these survival rates was

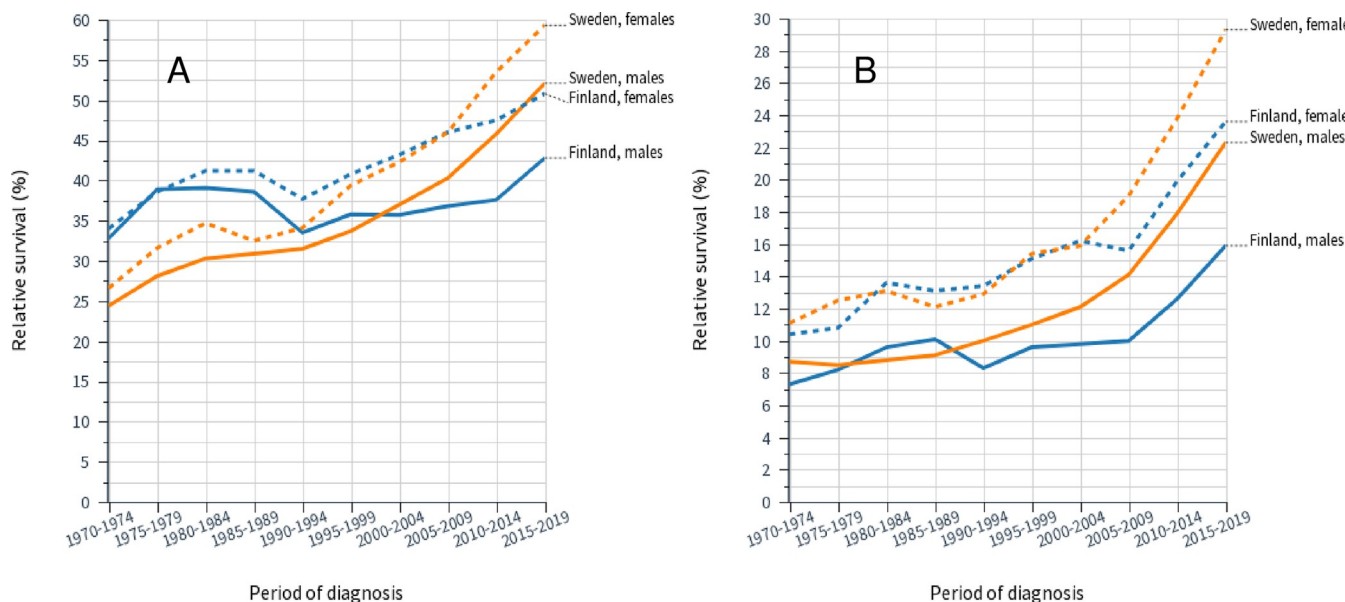

**Fig 5.** Relative 1-year (**A**) and 5-year (**B**) survival rates for lung cancer in Finnish and Swedish men and women. The related 95%CIs for the data points are shown in Table 3.

Table 3. Relative 1- and 5-year survival for lung cancer in Finland and Sweden.

| Period | 1-year survival % [95% CI] | | 5-year survival % [95% CI] | |
|---|---|---|---|---|
| Men | Finland | Sweden | Finland | Sweden |
| 1970–1974 | **32.8** [31.5–34.1] | 24.4 [23.3–25.5] | 7.3 [6.5–8.1] | 8.7 [7.9–9.5] |
| 1975–1979 | **38.9** [37.7–40.1] | 28.1 [26.9–29.3] | 8.2 [7.5–9.0] | 8.5 [7.8–9.3] |
| 1980–1984 | **39.1** [37.9–40.2] | 30.3 [29.2–31.5] | 9.6 [8.8–10.4] | 8.8 [8.1–9.6] |
| 1985–1989 | **38.6** [37.4–39.8] | 30.9 [29.8–32.1] | 10.1 [9.3–10.9] | 9.1 [8.4–9.9] |
| 1990–1994 | **33.5** [32.3–34.6] | 31.5 [30.4–32.7] | 8.3 [7.6–9.0] | 10.0 [9.3–10.8] |
| 1995–1999 | 35.8 [34.6–37.1] | 33.7 [32.5–34.9] | 9.6 [8.8–10.4] | 11.0 [10.2–11.9] |
| 2000–2004 | 35.7 [34.5–37.0] | 37.0 [35.9–38.2] | 9.8 [9.0–10.6] | **12.1** [11.3–13.0] |
| 2005–2009 | 36.8 [35.6–38.1] | **40.3** [39.1–41.5] | 10.0 [9.2–10.9] | **14.1** [13.2–15.1] |
| 2010–2014 | 37.6 [36.3–38.9] | **45.8** [44.6–47.0] | 12.6 [11.7–13.6] | **17.9** [16.9–18.9] |
| 2015–2019 | 42.8 [41.5–44.1] | **52.1** [50.8–53.3] | 15.9 [14.7–17.0] | **22.3** [21.1–23.5] |
| Women | Finland | Sweden | Finland | Sweden |
| 1970–1974 | **34.0** [30.6–37.4] | 26.6 [24.6–28.6] | 10.4 [8.1–13.1] | 11.1 [9.7–12.7] |
| 1975–1979 | **38.6** [35.6–41.6] | 31.6 [29.5–33.7] | 10.8 [8.9–12.9] | 12.5 [11.0–14.2] |
| 1980–1984 | **41.2** [38.5–43.9] | 34.7 [32.8–36.6] | 13.6 [11.6–15.7] | 13.1 [11.8–14.6] |
| 1985–1989 | **41.2** [38.8–43.7] | 32.5 [30.8–34.2] | 13.1 [11.4–15.0] | 12.1 [10.9–13.3] |
| 1990–1994 | 37.7 [35.4–39.9] | 34.1 [32.6–35.6] | 13.4 [11.7–15.1] | 12.9 [11.8–14.1] |
| 1995–1999 | 40.8 [38.8–42.9] | 39.4 [38.0–40.9] | 15.1 [13.5–16.7] | 15.4 [14.3–16.5] |
| 2000–2004 | 43.2 [41.3–45.2] | 42.3 [41.0–43.6] | 16.2 [14.7–17.8] | 15.9 [14.9–16.9] |
| 2005–2009 | 46.0 [44.2–47.8] | 46.0 [44.8–47.1] | 15.6 [14.2–17.0] | **19.0** [18.0–19.9] |
| 2010–2014 | 47.5 [45.7–49.2] | **53.5** [52.3–54.6] | 19.9 [18.5–21.4] | **23.8** [22.8–24.8] |
| 2015–2019 | 50.8 [49.1–52.4] | **59.3** [58.1–60.4] | 23.6 [22.0–25.2] | **29.3** [28.1–30.4] |

Bolding: 95%CI do not overlap. Bolding shows the country of higher survival %.

that there was no time dependent improvement, as all consecutive (and first and last) 95%CIs overlapped. Bolding in **Table 2** indicates significant difference between FI and SE, all of which were in favor of SE men.

Similar analysis of lung cancer survival is shown in **Table 3**. Because of larger case numbers than in laryngeal cancer many significant differences were observed, both between the countries and follow-up periods. FI men had initially a superior 1-year survival compared to SE men but in the last periods the survival advantage changed to SE. These shifts were almost identical for female 1-year survival. The differences were less in 5-year survival but the late SE advantage was evident for men and women. Time-dependent improvement was evidenced by non-overlapping 95%CI between first and last periods for all populations.

From 1970–74 to 2015–19 1-year survival for all cancer (not including non-melanoma skin cancer) increased for FI men from 48.3 to 80.7% and for FI women from 58.5 to 83.4%; for 5-year survival the percentages were 24.4 to 67.4 and 37.3 to 70.7, respectively. For SE men the increases for 1-year survival were from 55.7 to 87.3.7% and for SE women from 63.6 to 85.6%; for 5-year survival the percentages were 34.7 to 75.5 and 44.5 to 72.6, respectively.

## Discussion

The results describe incidence and survival tends for two smoking related cancers when the prevalence of smoking in the source populations decreased extensively. The present data agree with the individual-level epidemiological data that smoking related risk in laryngeal cancer decreases relatively faster than that for lung cancer [1]. The present data among FI men and SE

men and women also showed that the maximal incidence for laryngeal cancer occurred 5 to 10 years earlier compared to lung cancers. This was confirmed in birth cohort analysis of specific diagnostic age groups. The high historical incidence for laryngeal and lung cancers in FI men has been explained by the high prevalence of smoking origination from the wartime and post-war period. The war involved most FI men, and 5 cigarettes belonged to the daily ration of every solder [7]. Addicted smokers used additionally high-nicotine and high-tar Russia-type 'makhorka' (Nicotiana Rustica) which was grown in FI [10, 26]. Some authors have suggested that the FI habit of weekly or more frequent bathing in wood heated, smoky saunas may have contributed to lung cancer risk [27].

Sex difference in incidence was larger for laryngeal (20-fold in FI and 10-fold in SE) compared to lung cancer (15-fold in FI and 5-fold in SE). An interesting difference was observed in the trends of female cancers, lung cancer incidence was steadily increasing (5-fold in FI reaching no maximum and 7-fold in SE with a maximum in 2016) while laryngeal cancer incidence decreased in FI opposite to the increasing trend in SE. One possible explanation could be that among FI women risk factors other than smoking were important earlier, and the above reference to smoky saunas and in general wood-based cooking and heating may have contributed to FI female laryngeal cancers.

Against the background of extensive historical smoking differences between FI men compared to SE men, survival rates did not differ much in laryngeal cancer. FI women were initially disadvantaged (because of early mortality) but in the course of follow-up the differences disappeared [4]. Our overall conclusion about limited improvement in laryngeal cancer agrees with the Danish results [16]. The authors explained this by the historically favorable survival, thanks to early detections and low metastatic potential [16]. However, the 20% unit drop in survival between years 1 and 5 after diagnosis should be a challenge to novel treatments. While achievements in larynx-preserving techniques have increased the patients' quality of life they have not been equally successful in improving survival [4, 17]. The comparison of laryngeal cancer survival to all cancers underscores the disparity: while in the 1970s laryngeal cancer survival was much more favorable than that for all cancer, by 2015 all cancer had caught up in 1-years survival and passed laryngeal cancer in 5-year survival in FI and SE.

The above reference to historically good survival in laryngeal cancer does not apply to lung cancer for which survival has been dismal. We observed that heavy smoking FI men and light smoking FI women survived better during year 1 that their SE counterparts until the 1990s but the advantage shifted to SE men and women at around year 2010, and this was also evident for 5-year survival. The major improvement in recent 5-year survival for lung cancer is an important achievement. The most likely explanation is earlier diagnosis as more imaging and more sensitive modalities are increasingly used [28]. Treatment in an early stage lung cancer can be curative [29]. Why FI men were surviving worse than other groups would need special attention. These differences were not noted in an earlier study which reported data on 5-year survival only [4]. However, the authors also studied death rates up to 3 months of diagnoses and noted much lower mortality until about 1990 for FI men and women compared to their SE counterparts. The notion that there was no apparent correlation with laryngeal and lung cancer survival and population frequency of smoking can probably be rationalized by the fact that most patients were smokers in FI as well as in SE, due to high risk and high PAF of smoking [3].

The strength of the study data from two countries over a period of 50 years (and longer for incidence data) with practically free medical care offered to the population at large, covered by high-level nation-wide cancer registries of high-quality. Such circumstances are commensurate with the concept of 'real world' experience. The weaknesses are that the data are ecological and no individual level clinical and treatment data were available. Data on possible

confounders were lacking but as discussed in the first paragraph of Introduction smoking is the predominant risk factor for these cancers [30]. Smoking prevalence and prevalence of other relevant risk factors is well known in these populations (see the second paragraph of Introduction). Based on the US smoking prevalence data, cigarette consumption and lung cancer mortality showed a correlation of 0.91 when a latency of 18 or more years was applied [31]. Another weakness is that the SE cancer registry does not include cancers for which the only information is the death certificate [21]. As these cases tend to be relative more advanced and older than other cases, there omission may improve survival rates [4, 22].

In conclusion, incidence in laryngeal and lung cancers has drastically changed in FI and SE men due to reduced smoking prevalence. The female changes have been more moderate with a divergent recent increase in lung cancer and laryngeal cancer in SE women only. Laryngeal cancer survival, although relatively high has not improved compared to lung cancer with a low starting levels. Historical smoking prevalence appeared to be unrelated to survival trends between men and women, and FI and SE. As laryngeal and lung cancers are smoking related, the most efficient way to fight them is to promote non-smoking.

## Author Contributions

**Conceptualization:** Anni Koskinen, Kari Hemminki.

**Data curation:** Kari Hemminki.

**Formal analysis:** Otto Hemminki, Asta Försti, Kari Hemminki.

**Funding acquisition:** Kari Hemminki.

**Investigation:** Anni Koskinen, Kari Hemminki.

**Resources:** Otto Hemminki.

**Supervision:** Anni Koskinen, Asta Försti.

**Validation:** Otto Hemminki, Asta Försti, Kari Hemminki.

**Writing – original draft:** Anni Koskinen, Kari Hemminki.

**Writing – review & editing:** Otto Hemminki, Asta Försti.

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
