## [Decision Letter · Decision Letter 0]

9 Mar 2022

PONE-D-22-05948INCIDENCE AND SURVIVAL IN LARYNGEAL AND LUNG CANCERS IN FINLAND AND SWEDEN THROUGH A HALF CENTURYPLOS ONE

Dear Dr. Kari Hemminki,

Thank you for submitting your manuscript to PLOS ONE. After careful consideration, we feel that it has merit but does not fully meet PLOS ONE’s publication criteria as it currently stands. Therefore, we invite you to submit a revised version of the manuscript that addresses the points raised during the review process.

This study is generally reasonable with sufficient evidence but two reviewers have some concerns. Please revise this MS according to the suggestions or discuss them if the issues were not possible to address.

We look forward to receiving your revised manuscript.

Kind regards,

Wen-Wei Sung, M.D., Ph.D.

Academic Editor

PLOS ONE

Journal Requirements:

"no issues"

Reviewers' comments:

Reviewer's Responses to Questions

**Comments to the Author**

1. Is the manuscript technically sound, and do the data support the conclusions?

Reviewer #1: Yes

Reviewer #2: Yes

2. Has the statistical analysis been performed appropriately and rigorously? 

Reviewer #1: Yes

Reviewer #2: Yes

3. Have the authors made all data underlying the findings in their manuscript fully available?

Reviewer #1: Yes

Reviewer #2: Yes

4. Is the manuscript presented in an intelligible fashion and written in standard English?

Reviewer #1: Yes

Reviewer #2: Yes

5. Review Comments to the Author

Reviewer #1: I would like to mention the follwoing comments:

1- The comparability of data in two countries needs to be explained.

2- The potential confounders have not been considered.

3- The affecting factors on incidence have not been discussed.

4- Were APC (Age, Period, Cohort) applied for time trends?

5- The applied statistical package?

Good Luck

Reviewer #2: This study addressed the incidence and survival of laryngeal and lung cancers in Finland and Sweden through a half-century. The results are useful for public health. However, the following concerns should be addressed.

Abstract

1. A part of the abstract section should be dedicated to the methodology of the study. Furthermore, only the main results, regarding the objective of the study (incidence and survival), should be reported and other findings can be presented in the main text. The conclusion should be meaty and concise as well.

Introduction

2. The introduction is too long. Some parts can be removed such as the second half of the first paragraph.

3. The introduction section should include an overview of what is known on the topic, an explanation of what is unknown, and finally a description of what will be addressed by the study.

Methods

4. The statistical software used for analysis as well as the significance level should be specified.

Results

5. The total number of lung and laryngeal cases should be reported in the first paragraph of the main results.

Conclusion

6. Since the analyses are performed on ecologic data, no individual data on smoking status is available. Therefore, linking fluctuations of the incidence rate of lung and laryngeal cancers to the prevalence of smoking in this study does not make sense.

6. PLOS authors have the option to publish the peer review history of their article (what does this mean?). If published, this will include your full peer review and any attached files.

Reviewer #1: **Yes: **Masoud Amiri

Reviewer #2: No

---

## [Author Response · Author response to Decision Letter 0]

2 May 2022

Comments to the Author

5. Review Comments to the Author

Reviewer #1: I would like to mention the follwoing comments:

1- The comparability of data in two countries needs to be explained.

>>> We added text to the second sentence in Methods, p.4. Note that this theme is highlighted also in the last paragraph of Introduction, p. 4. 

2- The potential confounders have not been considered.

>>> Text added (to strengths/weaknesses paragraph), p. 7.

3- The affecting factors on incidence have not been discussed.

>>> The overwhelming influence of smoking is discussed in the above section and throughout, including the first paragraph if Introduction.

4- Were APC (Age, Period, Cohort) applied for time trends?

>>> We added text to methods, p.4, birth cohort analysis results to p. 5, and the new Fig. 2, and Discussion first paragraphs, p. 6.

5- The applied statistical package?

>>> Explained in Methods, paragraph 2, p. 4. 

Good Luck

Reviewer #2: This study addressed the incidence and survival of laryngeal and lung cancers in Finland and Sweden through a half-century. The results are useful for public health. However, the following concerns should be addressed.

Abstract

1. A part of the abstract section should be dedicated to the methodology of the study. Furthermore, only the main results, regarding the objective of the study (incidence and survival), should be reported and other findings can be presented in the main text. The conclusion should be meaty and concise as well.

>>> Abstract was modified accordingly.

Introduction

2. The introduction is too long. Some parts can be removed such as the second half of the first paragraph.

>>> The second half was deleted. Treatment should not be deleted in survival studies.

3. The introduction section should include an overview of what is known on the topic, an explanation of what is unknown, and finally a description of what will be addressed by the study.

>>> These points are discussed in paragraphs 2 and 3 of Introduction.

Methods

4. The statistical software used for analysis as well as the significance level should be specified.

>>> Explained in Methods, paragraph 2, p. 4.

Results

5. The total number of lung and laryngeal cases should be reported in the first paragraph of the main results.

>>> Done, p. 5.

Conclusion

6. Since the analyses are performed on ecologic data, no individual data on smoking status is available. Therefore, linking fluctuations of the incidence rate of lung and laryngeal cancers to the prevalence of smoking in this study does not make sense.

>>> When smoking is by far the overwhelming contributioning factor it is playing a role at the population level (see first paragraph of Introduction, p. 3 and the new text on Discussion, p. 7: correlation coefficient 0.91!)

---

## [Decision Letter · Decision Letter 1]

11 May 2022

INCIDENCE AND SURVIVAL IN LARYNGEAL AND LUNG CANCERS IN FINLAND AND SWEDEN THROUGH A HALF CENTURY

PONE-D-22-05948R1

Dear Dr. Kari Hemminki,

We’re pleased to inform you that your manuscript has been judged scientifically suitable for publication and will be formally accepted for publication once it meets all outstanding technical requirements.

Kind regards,

Wen-Wei Sung, M.D., Ph.D.

Academic Editor

PLOS ONE

Reviewers' comments:

Reviewer's Responses to Questions

**Comments to the Author**

1. If the authors have adequately addressed your comments raised in a previous round of review and you feel that this manuscript is now acceptable for publication, you may indicate that here to bypass the “Comments to the Author” section, enter your conflict of interest statement in the “Confidential to Editor” section, and submit your "Accept" recommendation.

Reviewer #1: All comments have been addressed

Reviewer #2: All comments have been addressed

2. Is the manuscript technically sound, and do the data support the conclusions?

Reviewer #1: Yes

Reviewer #2: Yes

3. Has the statistical analysis been performed appropriately and rigorously? 

Reviewer #1: Yes

Reviewer #2: Yes

4. Have the authors made all data underlying the findings in their manuscript fully available?

Reviewer #1: Yes

Reviewer #2: Yes

5. Is the manuscript presented in an intelligible fashion and written in standard English?

Reviewer #1: Yes

Reviewer #2: Yes

6. Review Comments to the Author

Reviewer #1: All my comments have been addressed. The authors answered my comments one by one, with clarifying the answers in the pages.

Reviewer #2: (No Response)

7. PLOS authors have the option to publish the peer review history of their article (what does this mean?). If published, this will include your full peer review and any attached files.

Reviewer #1: **Yes: **Masoud Amiri

Reviewer #2: No

---

## [Editor Report · Acceptance letter]

18 May 2022

PONE-D-22-05948R1 

INCIDENCE AND SURVIVAL IN LARYNGEAL AND LUNG CANCERS IN FINLAND AND SWEDEN THROUGH A HALF CENTURY 

Dear Dr. Hemminki:

I'm pleased to inform you that your manuscript has been deemed suitable for publication in PLOS ONE. Congratulations! Your manuscript is now with our production department. 

Kind regards, 

on behalf of

Dr. Wen-Wei Sung 

Academic Editor

PLOS ONE